# Burnout and sleep problems among nurses working in a tertiary hospital in Kathmandu, Nepal

**Manoj Panthi Kanak**●*, **Smriti Pant**

Institute of Medicine, Tribhuvan University, Maharajgunj, Kathmandu, Nepal

* manojpanthi@iom.edu.np

## Abstract

A growing number of professions are being affected by burnout but healthcare providers especially nurses are among the most affected. Due to the need to provide continuous medical care, nurses are a workforce that are obliged to engage in shift work causing sleep inadequacy and disturbance in regular sleep patterns in a working condition that is already physically and emotionally stressful. Burnout and sleep problems affects not just the well-being of nurses but also impacts their work efficiency resulting in reduced quality of care and safety of the patients. As there is very little existing information on this topic in Nepal, this study was conducted to determine the situation of burnout, sleep quality and, their correlates among nurses working in a tertiary hospital in Kathmandu, Nepal. This was a cross-sectional study in which quantitative method was applied. Data was collected from 246 nurses working in the Tribhuvan University Teaching Hospital using simple random sampling method between March and April, 2022. A self-administered questionnaire containing Oldenburg Burnout Inventory and Pittsburgh Sleep Quality Index was used to collect data and the collected data was analyzed using SPSS 26. More than three out of four nurses (78.5%) were found to have burnout, while 88.3% were disengaged and 83.0% were exhausted. Likewise, more than half of the nurses (58.9%) were found to have a poor sleep quality. Using multivariate logistics regression analysis, factors like position of the nurse (AOR = 6.0, 95% CI; 1.9-18.8, p-vale = 0.002) and, slight problem (AOR = 6.6, 95% CI; 3.0-14.7, p-value<0.001) and somewhat to a big problem (AOR = 6.3, 95% CI; 1.9-20.9, p-value = 0.002) of daytime dysfunction were found to be significantly associated with burnout. These results indicate the necessity to reduce burnout and manage sleep problems of the nurses prioritizing the nursing staffs occupying junior position. Similarly, establishment of interventions like psychological help desk and support groups for the nurses could be beneficial to mitigate the effects of burnout and sleep problems among the nurses.

**Data availability statement:** All relevant data are within the paper and its Supporting Information files.

**Funding:** The author(s) received no specific funding for this work.

**Competing interests:** The authors have declared that no competing interests exist.

## Author summary

Manoj Panthi Kanak is a young public health worker passionate about working at the intersection of LGBTQIA+ rights and public health. In the past few years, Kanak has engaged in numerous projects related to research, evidence generation, advocacy and program implementation in the field of LGBTQIA+ health. Their work primarily focuses on mental health, HIV prevention, harm reduction, sexual and reproductive health and rights, and human rights advocacy for LGBTQIA+ community including other underrepresented and marginalized groups in Nepal. They have a Bachelor degree in Public Health from Institute of Medicine, Tribhuvan University, Maharajgunj, Kathmandu, Nepal. Kanak performed this study with the help of their college supervisor as a part of their undergraduate research project for Bachelor of Public Health with the aim to determine the situation of burnout, sleep quality and, their correlates among nurses working in Tribhuvan University Teaching Hospital, Institute of Medicine, Maharajgunj, Kathmandu, Nepal.

## Introduction

Nurses are a workforce who have to work in extended shift hours to provide continuous medical care for 24 hours every day in a work environment which is demanding and full of physical and emotional stressors [1,2]. Studies show that stress is a factor that affects nurses on a daily basis and can result in nurses' absenteeism and aggression as well as reduced productivity and efficiency, diminishing the quality of care and patient safety [3,4]. Healthcare profession is already a highly hazardous occupation and studies suggest that nurses facing burnout are more likely to provide suboptimal care, compromise patient safety by poorer prescription or increased work accidents [5–7]. Shift work has been found to interfere with the circadian and homeostatic regulation of sleep of the nurses causing most nurses working night shifts to struggle adjusting to daytime activities or normal night sleep patterns on their days off [4].

Burnout is defined as a sustained response to chronic work-related stresses and is often measured in three dimensions, namely: a) emotional exhaustion; experience of being emotionally exhausted, b) depersonalization or disengagement; establishment of detached, distant, and cynical relationships with patients and colleagues and, c) feeling of low personal accomplishment and professional failure [8,9]. With increasing work-culture and high demanding jobs, burnout affects workers in a growing number of professions and nurses including other healthcare providers are among the most often affected [8]. Nursing is an occupational area presenting one of the highest levels of work-related stress and burnout, which has been associated to various occupational and personal factors [8,10]. Burnout and sleep problems has been closely associated among healthcare providers and this relation has been confirmed in many observational studies among other types of workers as well [10].

A good sleep is vital for maintaining good health as it affects hormonal levels, mood and physiology. Besides the role of sleep in learning, memory consolidation and motor learning, essential in any professional domain, it has a key role in emotional regulation as well [10]. Complaints of fatigue and insufficient or poor quality of sleep is common among nurses working in 12 hours shifts. Nurses working successively on 12-hours shifts were found to have poor physical and cognitive recovery due to inadequate amount of sleep between the shifts, in a study from the United States [11]. Association between dissatisfaction with sleep patterns and emotional exhaustion among nurses, and a high level of depersonalization among those working the day shift was found in a Brazilian study [1]. Insufficient sleep is a potential cause of burnout, explored in a study among professionals from an information-technology company where less than six hours of sleep at night was the main risk factor for developing burnout [12].

Between 57% and 83.2% of shift nurses worldwide report sleep problems, including sleep disturbances, sleep deprivation, and poor sleep quality [4]. The global prevalence of emotional exhaustion and depersonalization among nurses is 34.1% and 12.6%, respectively, according to a metanalytical study from 2021 [13]. In a study from 2018 among Asian countries including Nepal, the level of burnout among nurses working in intensive care units was found to be 52.0%, $P = 0.362$ [9]. Recent data from research among nurses in India conducted after the second wave of COVID-19 pandemic shows burnout among 66% of the respondents [14]. Studies from Nepal have shown 65.9% and 38.2% prevalence of burnout and poor sleep quality among medical students, respectively [15,16]. However, burnout and sleep problems of nurses has not been adequately explored in Nepal. This study was carried out to fill this knowledge gap by determining the situation of burnout, sleep quality and, their correlates among nurses working in a tertiary hospital in Kathmandu, Nepal.

## Materials and methods

### Ethics statement

Ethical approval was obtained from Institutional Review Committee – Institute of Medicine (IOM) (Reference number: 339(6–11)E²078/079) and approval to carry out the research was taken from Tribhuvan University Teaching Hospital (TUTH) administration with coordination from Central Department of Public Health, IOM. The research was conducted with the written informed consent of the participants and their voluntary enrollment after thoroughly informing about the research objectives and their responsibilities as research respondents. Confidentiality of the data collected was ensured through coding of each response and the respondents were kept anonymous.

### Study design

A cross-sectional study was conducted among the nurses working in the TUTH, IOM, Maharajgunj, Nepal. TUTH is one of the oldest and largest tertiary government hospitals in Nepal which provides specialized services in departments ranging from urology, neurosurgery, psychiatrics, dermatology, pediatrics, to name a few. With its comprehensive services and location in the capital city, TUTH is a referral site for district and provincial hospitals from all across Nepal. TUTH was selected as study site because of its high patient load and an adequate sample population of nurses. Data collection was done among 246 nurses working in the TUTH between March and April, 2022.

### Study sample

The study followed the simple random sampling method, using computerized number generator without replacement. The details of the study population were obtained from the administrative section of the hospital. A list of the nurses was developed with each unit of these total study population assigned a unique id, and computerized system using Ms-Excel was used to pick the required sample size from the list. Those selected were noted and contacted through their supervisors, and after receiving their written consent the questionnaire was distributed.

The sample size was determined using the Cochran's formula for sample size collection from Sampling Techniques as stated in equation 1 [17].

$$n = (Z^2 \times p \times q)/e^2$$

<div align="right">equation 1</div>

Where

Z = standard normal deviation, usually set at 1.96 which corresponds to 95% confidence level
p = proportion in the target population estimated = 31% [8]
e = degree of accuracy required, usually set at 0.05 level
n = desired sample size which was calculated to be 328
For finite population N = 805 adjusted sample size was calculated to be 233 using equation 2

$$\textit{Adjusted sample size} = n/\big[1 + \{(n-1)/N\}\big]$$

<div align="right">equation 2</div>

Again, with 10% of non-response rate the final sample size was determined to be 256

Questionnaires were distributed starting 20/03/2022 and were collected within 16/04/2022. Data was entered and processed subsequently after collection.

## Tools for data collection

The study used self-administered questionnaire with three sections; a) socio-demographic and work-related questions, followed by assessment of b) burnout and c) sleep quality. The first section of the questionnaire was used to collect personal and work-related data. It included information regarding the age, education level, marital status, health status, religiosity, cigarette use, alcohol use, exercise, work department and, the number of nurses and beds in the department. Sex was not included in the questionnaire after confirmation from the administration that all the nurses working at TUTH were female.

Although Maslach Burnout Inventory is the most commonly used scale to measure burnout, it's third factor personal accomplishment has been shown to perform weakly [18]. Oldenburg Burnout Inventory (OLBI-S) which contains 16 items questionnaire focusing on the two dimensions of burnout; disengagement and exhaustion was thus used to assess burnout among the nurses in this study [19]. OLBI-S subscales consist of eight items each, among which four are worded positively and four are worded negatively. Participant's responses were collected on a four-point Likert scale ranging from 1 (strongly agree) to 4 (strongly disagree). Scores from the negatively worded eight items were reversed so that higher score corresponds to higher burnout level.

Pittsburgh Sleep Quality Index (PSQI) was used to assess sleep quality through its seven parameters namely: subjective sleep quality, sleep latency, sleep duration, habitual sleep efficiency, sleep disturbances, use of sleep medication and day-time dysfunction. Each of the seven component has a score between 0 (no problem) to 3 (severe problem) which is added to get the global PSQI score. Both the tools OLBI-S and PSQI have been previously used in Nepal in similar settings [15,16].

## Analytical strategy

The scores from OLBI-S were categorized for prevalence of disengagement and exhaustion using the cutoff of mean ≥ 2.10 from disengagement subscale and mean ≥ 2.25 from exhaustion subscale [20]. Respondents who were found to be both disengaged and exhausted were termed 'Burnout' as shown in Fig 1 and the rest were termed 'No burnout' for bivariate analysis.

Higher score in PSQI indicate poorer sleep quality, and a global score of 5 has been established as the cutoff thus, those with score five or more were labelled to have a 'poor sleep quality' while the rest were labelled 'good sleep quality' [21].

**PLOS Global Public Health**

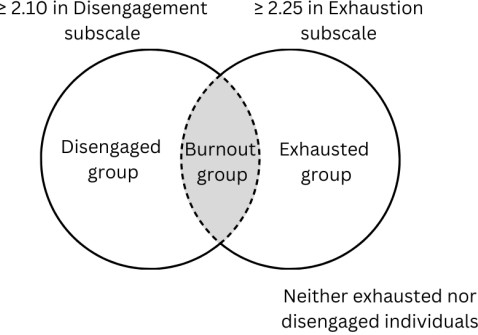

≥ 2.10 in Disengagement subscale ≥ 2.25 in Exhaustion subscale

Disengaged group Burnout group Exhausted group

Neither exhausted nor disengaged individuals

**Fig 1. Classification of burnout using OLBI-S.**

The collected data was entered in EpiData v3.1 and was exported to SPSS 26 for analysis. Burnout was the dependent variable while the socio-demographic and work-related characteristics including the seven parameters of the sleep quality obtained from PSQI were the independent variables. Descriptive analysis of the variables was done in terms of frequencies and percentage. Chi-square test for categorical independent variables was carried out with burnout as an outcome of interest (with 1 = yes and 0 = no). Variables with a p-value less than 0.1 during bivariate analysis were fitted into the multivariate regression model [22]. The adjusted odds ratio was calculated at a 95% CI, and a p-value less than 0.05 was considered statistically significant. The value of Cronbach's Alpha for 16 items of OLBI-S was 0.787 indicating a good level of internal consistency.

## Results

Out of the 256 self-administered questionnaires distributed, a total of 246 were collected from the nurses working at TUTH indicating a response rate of 96%. Among the study participants, a very large number of the respondents (78.5%) were found to have burnout while more than half (58.9%) reported poor sleep quality.

### Socio-demographic characteristics of the study participants

The age of the 246 study participants was distributed from 20 to 59 years with mean age of 30 years. Of the study participants, 80.5% had a Bachelor degree (Bachelor of Science in Nursing or Bachelor in Nursing), more than half (56.5%) were married and 89.4% were religious. In the last month from the data collection, 58.5% had not exercised, 91.9% had not consumed alcohol and 99.2% had not consumed cigarette (Table 1).

### Work-related characteristics of the study participants

Among the study participants, staff nurses were the largest represented (92.7%) and a large proportion of the nurses (58.12%) were stationed at general wards. Similarly, 46.3% of the respondents had a work experience of less than 5 years with the mean work experience of 8.20 years and the mean beds to nurses ratio was 1.4 ± 0.86, which meant for every five nurses there were an average of seven beds in the hospital (Table 2).

### Magnitude of burnout and sleep problems among the study participants

Among the respondents, 88.2% scored ≥2.10 in disengagement subscale and 82.9% scored ≥2.25 in exhaustion subscale of OLBI-S. Nurses who were both disengaged and exhausted i.e., those with burnout were found to be 78.5%. Similarly, the mean PSQI global score of the respondents was 5.43 with SD of ±3.035. A PSQI global score of ≥5 was found among 58.9% implying poor sleep quality.

**Table 1. Socio-demographic charactersitics of the study participants.**

| Characteristics | Number | N = 246 Percentage |
|---|---|---|
| Age (years) | | |
| Mean | 30 | |
| SD | ±7.51 | |
| Highest education | | |
| Diploma/PCL Nursing | 37 | 15.0 |
| Bachelors | 198 | 80.5 |
| Masters | 11 | 4.5 |
| Marital status | | |
| Married | 139 | 56.5 |
| Unmarried | 106 | 43.1 |
| Divorced | 1 | 0.4 |
| Religious | | |
| Yes | 220 | 89.4 |
| No | 26 | 10.6 |
| Exercise | | |
| Not during the past month | 114 | 58.5 |
| Less than once a week | 38 | 15.4 |
| Once or twice a week | 36 | 14.6 |
| Three or more times a week | 28 | 11.4 |
| Alcohol consumption | | |
| Not during the past month | 226 | 91.9 |
| Less than once a week | 16 | 6.5 |
| Once or twice a week | 3 | 1.2 |
| Three or more times a week | 1 | 0.4 |
| Cigarette consumption | | |
| Not during the past month | 244 | 99.2 |
| Less than once a week | 2 | 0.8 |

In the bivariate analysis, position of the nurses including subjective sleep quality, sleep disturbance and daytime dysfunction measured from PSQI were found to be significantly associated with burnout among the study participants (Table 3). No significant association of burnout was found with the age, level of education, work experience, marital status, religiousness, engagement in physical exercise, alcohol consumption, including other parameters of sleep quality.

### Factors associated with burnout

The odds of burnout among study participants who were working in the position of staff nurse were 6.0 (95% CI; 1.9-18.8, p-value = 0.002) times higher than those working in the position of nursing officer or supervisor. In regards to daytime dysfunction, study participants who reported a slight problem were 6.6 (95% CI; 3.0-14.7, p-value<0.001) times and who reported somewhat to a big problem were 6.3 (95% CI; 1.9-20.9, p-value = 0.002) times more likely to have burnout than those who reported no any problem of day time dysfunction (Table 4).

### Discussion

This study aimed to find the prevalence of burnout and sleep problems among nurses, and the associated factors of burnout with various socio-demographic and work-related characteristics. The prevalence of burnout among the nurses in this

**Table 2. Work related characteristics of the study participants.**

| Characteristics | Number | N = 246 Percentage |
|---|---|---|
| Position | | |
| Staff Nurse | 228 | 92.7 |
| Nursing Officer | 15 | 6.1 |
| Nursing Supervisor | 3 | 1.2 |
| Work experience (years) | | |
| Mean | 8.20 | |
| SD | ±7.74 | |
| Department | | |
| General wards | 143 | 58.12 |
| Intensive Care Units | 59 | 23.98 |
| Operation Theatre | 16 | 6.50 |
| Emergency | 12 | 4.87 |
| Psychiatry wards | 11 | 4.47 |
| Out Patient Departments | 5 | 2.03 |
| Beds to nurses ratio | | N = 224 |
| Mean | 1.40 | |
| SD | ±0.86 | |

study was 78.5% while, 88.2% were disengaged and 82.9% were exhausted. The prevalence of burnout was found to be much higher than the literature around the globe [1–3,10,23]. Similar to the findings of this study, a study among health-care professionals in Nepal from 2021 showed that 92.2% and 3.9% of nurses have moderate and high level of burnout, respectively [24]. However, according to a multinational study that included Nepal the prevalence of burnout among nurses working in intensive care units was 52%, much lower than this study [9]. Similarly, a study among undergraduate medical students in Nepal report prevalence of burnout among 65.9% [15]. Since this study was done post the COVID-19 pandemic the results could have been exacerbated as suggested in a study where pandemic fatigue was found to have a significant negative correlation with mental health, job contentment and sleep quality among the nurses [25]. In this study, prevalence of disengagement was found to be higher than exhaustion which was different to the result from a metanalytic study which reported emotional exhaustion to be the most common dimension of burnout [8].

This study revealed a poor sleep quality among the nurses (58.9%). Although there is no previous data regarding sleep quality among nurses in Nepal, existing evidence suggest poor sleep quality among healthcare providers in a private hospital (48.03%) and undergraduate medical students (38.2%) [16,26]. The mean PSQI score of the nurses in this study was 5.43±3.035, which was lesser in comparison to some studies from around the globe indicating a better sleep quality [2,27–29]. According to a study, nurses have an average sleep of less than 6 hours and this pattern of short sleep could be due to a lack of sleep opportunity rather than sleep ability [11]. In this study however, the mean sleep hours of the nurses was 6.8±1.1 hours, which is not very less than the recommended seven to eight hours of daily sleep for adults [10].

This study demonstrates a significant association between the position and burnout among the nurses. Other studies also suggest that nurses working in lower positions have greater odds of burnout than their supervisors [1,3,30]. This may be due to greater work load among staff nurses or delegation of work from their supervisors, while lower control over the work. A study suggests that higher level nurses in comparison to their subordinates can better cope high-demanding situations due to greater control over work [31]. Similarly, lack of promotion opportunity and unfair evaluation of work among the nurses could be a barrier to their work environment and health, which could be accounted to greater burnout among the staff nurses [32].

PLOS Global Public Health

**Table 3. Association between burnout and categorical independent variables among the study participants.**

| Characteristics | Burnout (%) | No burnout (%) | Total (%) | Chi- square value | p-value |
|---|---|---|---|---|---|
| Age | | | | | |
| Less than/ equal to 30 | 131 (80.4) | 32 (19.6) | 163 (66.3) | 1.0 | 0.306 |
| Greater than 30 | 62 (74.7) | 21 (25.3) | 83 (33.7) | | |
| Education | | | | | |
| Bachelors/Masters | 165 (78.9) | 44 (21.1) | 209 (84.9) | 0.2 | 0.655 |
| Diploma | 28 (75.7) | 9 (24.3) | 37 (15.1) | | |
| Position | | | | | |
| Staff Nurse | 186 (81.6) | 42 (18.4) | 228 (92.7) | 17.9 | <0.001 |
| Nursing Officer/ Supervisor | 7 (38.9) | 11 (61.1) | 18 (7.3) | | |
| Work experience | | | | | |
| Less than/ equal to 8 years | 128 (80.5) | 31 (19.5) | 159 (64.6) | 1.1 | 0.291 |
| Greater than 8 years | 65 (74.7) | 22 (25.3) | 87 (35.4) | | |
| Marital status | | | | | |
| Unmarried/ Divorced | 89 (83.2) | 18 (16.8) | 107 (43.5) | 2.5 | 0.114 |
| Married | 104 (74.8) | 35 (25.2) | 139 (56.5) | | |
| Religiousness | | | | | |
| Not religious | 22 (84.6) | 4 (15.4) | 26 (10.6) | 0.6 | 0.419 |
| Religious | 171 (77.7) | 49 (22.3) | 220 (89.4) | | |
| Physical exercise | | | | | |
| Never | 116 (80.6) | 28 (19.4) | 144 (58.5) | 0.9 | 0.341 |
| At least once a week | 77 (75.5) | 25 (24.5) | 102 (41.5) | | |
| Alcohol consumption | | | | | |
| Never | 175 (77.4) | 51 (22.6) | 226 (91.9) | 1.7 | 0.190 |
| At least once a week | 18 (90.0) | 2 (20.0) | 20 (8.1) | | |
| Subjective sleep quality | | | | | |
| Very good | 35 (63.6) | 20 (36.4) | 55 (22.4) | 13.2 | 0.001 |
| Fairly good | 122 (79.7) | 31 (20.3) | 153 (62.2) | | |
| Fairly and very bad | 36 (94.7) | 2 (5.3) | 38 (15.4) | | |
| Sleep latency | | | | | |
| ≤15 minutes | 47 (71.2) | 19 (28.8) | 66 (26.8) | 4.0 | 0.262 |
| 16-30 minutes | 79 (78.2) | 22 (21.8) | 101 (41.1) | | |
| 31-60 minutes | 42 (84.0) | 8 (16.0) | 50 (20.3) | | |
| >60 minutes | 25 (86.2) | 4 (13.8) | 29 (11.8) | | |
| Sleep duration | | | | | |
| ≥7 hours | 106 (77.4) | 31 (22.6) | 137 (55.7) | 4.3 | 0.114 |
| 6-7 hours | 60 (75.0) | 20 (25.0) | 80 (32.6) | | |
| <6 hours | 27 (93.1) | 2 (6.9) | 29 (11.7) | | |
| Sleep efficiency | | | | | |
| ≥85% | 119 (78.8) | 32 (21.2) | 151 (61.4) | 2.7 | 0.256 |
| 75-84% | 50 (73.5) | 18 (26.5) | 68 (27.6) | | |
| <75% | 24 (88.9) | 3 (11.1) | 27 (11.0) | | |
| Sleep disturbance | | | | | |
| Not during the past month | 12 (57.1) | 9 (42.9) | 21 (8.5) | 7.4 | 0.024 |
| Less than once a week | 152 (79.2) | 40 (20.8) | 192 (78.1) | | |
| Once or more a week | 29 (87.9) | 4 (12.1) | 33 (13.4) | | |

*(Continued)*

**Table 3.** (Continued)

| Characteristics | Burnout (%) | No burnout (%) | Total (%) | Chi- square value | p-value |
|---|---|---|---|---|---|
| Use of sleep medication | | | | | |
| Not during the past month | 179 (78.2) | 50 (21.8) | 229 (93.1) | 0.2 | 0.915 |
| Less than once a week | 9 (81.8) | 2 (18.2) | 11 (4.5) | | |
| Once or more a week | 5 (83.3) | 1 (16.7) | 6 (2.4) | | |
| Daytime dysfunction | | | | | |
| No problem at all | 33 (49.3) | 34 (50.7) | 67 (27.2) | 46.5 | <0.001 |
| Only a very slight problem | 109 (88.6) | 14 (11.4) | 123 (50.0) | | |
| Somewhat to a big problem | 51 (91.1) | 5 (8.9) | 56 (22.8) | | |

**Table 4.** Logistics regression analysis showing association between explanatory variables and burnout among the study participants.

| Explanatory variables | Burnout (%) | | Crude Odds Ratio (95% CI) | p-value | Adjusted Odds Ratio (95% CI) | p-value |
|---|---|---|---|---|---|---|
| | Yes (n = 193) | No (n = 53) | | | | |
| Position | | | | | | |
| Staff nurse | 186 (81.6) | 42 (18.4) | 6.9 (2.5-19.0) | <0.001 | 6.0 (1.9-18.8) | 0.002 |
| Nursing officer/supervisor | 7 (38.9) | 11 (61.1) | Ref | | Ref | |
| Subjective sleep quality | | | | | | |
| Very good | 35 (63.6) | 20 (36.4) | Ref | | Ref | |
| Fairly good | 122 (79.7) | 31 (20.3) | 2.5 (1.1-4.4) | 0.019 | 1.3 (0.6-2.8) | 0.557 |
| Fairly and very bad | 36 (94.7) | 2 (5.3) | 10.2 (2.2-47.3) | 0.003 | 3.2 (0.6-17.6) | 0.189 |
| Sleep disturbance | | | | | | |
| Not during the past month | 12 (57.1) | 9 (42.9) | Ref | | Ref | |
| Less than once a week | 152 (79.2) | 40 (20.8) | 2.8 (1.1-7.2) | 0.691 | 1.5 (0.5-4.5) | 0.498 |
| Once or more a week | 29 (87.9) | 4 (12.1) | 5.4 (1.4-21.1) | 0.071 | 1.4 (0.3-7.4) | 0.693 |
| Daytime dysfunction | | | | | | |
| No problem at all | 33 (49.3) | 34 (50.7) | Ref | | Ref | |
| Only a very slight problem | 109 (88.6) | 14 (11.4) | 8.0 (3.8-16.7) | <0.001 | 6.6 (3.0-14.7) | <0.001 |
| Somewhat to a big problem | 51 (91.1) | 5 (8.9) | 10.5 (3.7-29.6) | <0.001 | 6.3 (1.9-20.9) | 0.002 |

Studies around the globe suggest association between burnout and sleep problem [1,11,33]. Among the seven parameters within sleep quality, daytime dysfunction refers to trouble staying awake and lack of enthusiasm while carrying out daily activities. This study revealed a significant association between burnout and daytime dysfunction among the nurses. Daytime dysfunction could occur due to the demanding nature of shift work causing fatigue in the body and emotional stress among the nurses [1,34].

No significant association was seen between burnout among the nurses and, their age and work experience in this study. Some studies show that younger nurses are at greater risk of burnout, while others reveal that older nurses are more vulnerable to burnout. Studies that suggest that longer work experience makes nurses protected to burnout reveal this could be due to familiarization with the scope of work and the work environment [1,3,8]. A study suggests that with higher beds-to-nurses ratio there is increase in the workload, and subsequently the work stress of nurses [35]. A difference among emotional exhaustion between nurses working in acute medicine and emergency and accidents departments was noted in another study [8,36]. In this study however, significant association between work department and burnout among the nurses was not observed.

Although many literatures have associated burnout among nurses with higher education i.e., Bachelor or above in comparison to those with Diploma, no such association could be found in this study [9]. Married or living with a spouse and being religious have been linked with protective effect among nurses [3,9]. Being religious is believed to strengthen

people when coping with stress and work problems, and it often reduces the negative impact of these on mental health [37]. Likewise, physical exercise has been shown to have a protective effect against burnout and studies report that with greater intensity of physical exercises risk of burnout is lower [38]. However, no such association was found in this study between burnout among the nurses and their marital status, religiousness or physical exercise.

## Conclusion

A very high prevalence of burnout and poor sleep quality was found among the nurses in this study. Position and daytime dysfunction were significant correlates of burnout among the nurses. Based on the findings of this study, it is recommended to reduce work stress and manage sleep problems of the nurses especially focusing the junior position nursing staffs. There is a need to evaluate the burden of work at various departments and develop a suitable environment for the nurses to get adequate rest in between their shifts. Relevant strategies to enhance the mental health of the nurses through interventions like psychological help desk and support groups need to be established at the institutional level. Further prospective studies are required to better understand the cause-effect relationships of risk factors of burnout.

## Supporting information

**S1 Data. Data underlying the results of the study.** https://docs.google.com/spreadsheets/d/1o8st-4gHzoC8fNbLonhqJIMJgUEx69uc/edit?usp=drive_link&ouid=117602478348588754759&rtpof=true&sd=true.
(XLSX)

## Acknowledgement

We thank all the respondents; the staff nurses, nursing officers and supervisors for their cooperation, valuable participation and honest response during data collection. We are grateful to Tribhuvan University Teaching Hospital for providing administrative approval to conduct the study.

## Author contributions

**Conceptualization:** Manoj Panthi Kanak, Smriti Pant.

**Data curation:** Manoj Panthi Kanak.

**Formal analysis:** Manoj Panthi Kanak.

**Methodology:** Manoj Panthi Kanak, Smriti Pant.

**Project administration:** Manoj Panthi Kanak.

**Supervision:** Manoj Panthi Kanak, Smriti Pant.

**Validation:** Manoj Panthi Kanak.

**Visualization:** Manoj Panthi Kanak.

**Writing – original draft:** Manoj Panthi Kanak.

**Writing – review & editing:** Manoj Panthi Kanak, Smriti Pant.

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
