## [Decision Letter · Decision Letter 0]

PGPH-D-24-02324

Burnout and sleep problems among nurses working in a tertiary hospital in Kathmandu, Nepal

Dear Dr. Kanak,

Thank you for submitting your manuscript to PLOS Global Public Health. After careful consideration, we feel that it has merit but does not fully meet PLOS Global Public Health’s publication criteria as it currently stands. Therefore, we invite you to submit a revised version of the manuscript that addresses the points raised during the review process.

We look forward to receiving your revised manuscript.

Kind regards,

Evangelos C. Fradelos

Academic Editor

Journal Requirements:

1. Please provide a complete Data Availability Statement in the submission form, ensuring you include all necessary access information or a reason for why you are unable to make your data freely accessible. If your research concerns only data provided within your submission, please write "All data are in the manuscript and/or supporting information files" as your Data Availability Statement.

2. Please provide separate figure files in .tif or .eps format.

3. Please provide an Author Summary. This should appear in your manuscript between the Abstract (if applicable) and the Introduction, and should be 150–200 words long. The aim should be to make your findings accessible to a wide audience that includes both scientists and non-scientists. Sample summaries can be found on our website under Submission Guidelines: 

https://journals.plos.org/globalpublichealth/s/submission-guidelines#loc-parts-of-a-submission

Additional Editor Comments (if provided):

Reviewers' comments:

Reviewer's Responses to Questions

**Comments to the Author**

1. Does this manuscript meet PLOS Global Public Health’s publication criteria? Is the manuscript technically sound, and do the data support the conclusions? The manuscript must describe methodologically and ethically rigorous research with conclusions that are appropriately drawn based on the data presented.

Reviewer #1: Partly

Reviewer #2: Yes

2. Has the statistical analysis been performed appropriately and rigorously?

Reviewer #1: Yes

Reviewer #2: Yes

3. Have the authors made all data underlying the findings in their manuscript fully available (please refer to the Data Availability Statement at the start of the manuscript PDF file)?

Reviewer #1: Yes

Reviewer #2: Yes

4. Is the manuscript presented in an intelligible fashion and written in standard English?

Reviewer #1: No

Reviewer #2: Yes

5. Review Comments to the Author

Reviewer #1: Title: The study has public health implications for the mental health of nurses and clinical relevance for nursing practice. Findings from this study could be utilized to improve nurse efficiency and quality patient care

Abstract: line 13 - 34

• Introduction: The introduction lacks focus and is not justified. Terms such as "burnout" and "sleep problems" (e.g., sleep deprivation, disrupted circadian rhythms) are commonly used terms in nursing practice and do not require definition here. For the introduction:

Hint: Can we know, “To what extent do sleep problems and burnout affect nursing practice or patient care?” or what is the rationale for assessing the prevalence of BURNOUT among nurses? What is the gap?

• Objective: The study objective is missing. Clearly state the aim of the research in the abstract.

• Method: This section is not concise. What are the data collection tools for this study? These could be modified: Oldenburg Burnout Inventory and Pittsburgh Sleep Quality Index to capture the background characteristics of study participants. Indicate how the study variables were measured. How did you analyze the data? Did you use a statistical package for data analysis?

• Results: Ensure that the presentation of results aligns with standards for the logistic regression output (e.g., OR, 95% CI, p-value). Also, what for: multivariate analysis and independent T-test? T-test for what?

• Conclusion: The conclusion is generally well-written, but the author needs to be consistent with the use of terminologies (e.g., use either "burnout" or "work stress" consistently).

Main Work:

Background: If relevant, you could start the background from a global perspective, then move to Asia, India, Nepal, and specific regions like Kathmandu, This helps contextualize the findings of your study. This helps clarify whether burnout and sleep deprivation are widespread in nursing practice or unique to this setting.

Example: "Nurses often work 8–12-hour shifts. Line 36: Nurses are a workforce who have to provide continuous medical care 24 hours a day”. NOT CLEAR, how?

Refer to the do’s and don’ts of scientific writing. From lines 67-70, what was it meant to communicate? go straight. The identified gap in the study did not come out clear. What is the problem in that hospital that the study seeks to solve? The study objective was not found in the background and the study was not justified. Line 172: IS it a correlation analysis or a Chi-square analysis? Use the appropriate epidemiological term. From lines 173-183: the presentation was not consistent with Table 4. The p-values indicated in the write-up cannot be traced in the table. Also, Table 4 was an incomplete presentation of adjusted and unadjusted ORs. Where are their p-values? From this: p-value less than 0.001? This cannot be found in the method section.

Materials and methods

Be consistent: what was the study design? Where was the study conducted? This is a tertiary hospital that every person would wish to know the background. Tell us how nurses are burdened in that hospital e.g patient to nurse ratio. What informed the sampling technique and why not systematic sampling? how did you use the computerized systems for selecting the participants? The study variables were not well-defined in terms of independent and dependent variables.

Line 132-139: The analytical strategy was poorly described. Descriptive statistics for only background characteristics? Why did you do binary logistics regression and sample T-test? For which of the variables?

Results: Line 150-186: Tables are poorly presented. The descriptions of the tables were poorly done. Where is the output of the t-test?

General comments: Ensure the study aligns with scientific writing standards, presenting a clear and logical flow. Sentence construction and grammar should support clarity and readability.

Reviewer #2: INTRODUCTION

- Nicely written introduction. However, why should we care if nurses' burnout is not directly affecting the care recipients (patients)? Anyone can experience burnout. It would be helpful to emphasize how stress and burnout among nurses have affected the care provided to patients. For instance, in line 46 & 47 you mentioned burnout resulting to ".....professional failures". What are these failures? Are there reported statistics on how nurses may have failed to act appropriately due to stress or burnout, particularly in Nepal or countries with similar demographics? I recommend this addition to the already nicely written introduction.

- There seem to be a grammatical problem in line 68-71

RESULTS

- The results have been nicely presented. I suggest that you briefly present the prevalence of burnout in one sentence before introducing the socio-demographics variables, as this is a crucial finding that readers will be eager to see. Currently, it appears far down in the presentation. You may also include the sample size and response rate in one sentence before presenting the prevalence of burnout.

- Did you by any means thought of considering the association between gender/sex or marital status and experiencing burnout? some studies elsewhere have reported fascinating results on these variables.

DISCUSSION

I have read some recent and very similar studies that report burnout among nurses in Nepal. Look and see how helpful this can be in this paper. See

(1) Shrestha et al., (2021), available at: https://nepjol.info/index.php/IJOSH/article/view/37259

(2) Shah et al., (2024), available at: https://pmc.ncbi.nlm.nih.gov/articles/PMC11261546/

This can prompt you to look again into the literature to avoid duplication of results.

RECOMMENDATIONS

To the best of my knowledge, I think it would have been necessary to provide some recommendations to improve nurses’ efficiency and reduce stress and burnout in the workplace. It might not necessarily be a government policy recommendation, but an institutional recommendation that can be incorporated within the Bylaws of the institution or hospital or healthcare organizations.

6. PLOS authors have the option to publish the peer review history of their article (what does this mean?). If published, this will include your full peer review and any attached files.

**Do you want your identity to be public for this peer review?** For information about this choice, including consent withdrawal, please see our Privacy Policy.

Reviewer #1: No

Reviewer #2: No

---

## [Decision Letter · Decision Letter 1]

PGPH-D-24-02324R1

Burnout and sleep problems among nurses working in a tertiary hospital in Kathmandu, Nepal

Dear Dr. Kanak,

Thank you for submitting your manuscript to PLOS Global Public Health. After careful consideration, we feel that it has merit but does not fully meet PLOS Global Public Health’s publication criteria as it currently stands. Therefore, we invite you to submit a revised version of the manuscript that addresses the points raised during the review process.

The reviewers have assessed the revised manuscript and provided some further comments to improve the manuscript. Please review their comments below and make the appropriate revisions to address their concerns.

We look forward to receiving your revised manuscript.

Kind regards,

Emma Campbell, Ph.D

Staff Editor

Journal Requirements:

Additional Editor Comments (if provided):

Reviewers' comments:

Reviewer's Responses to Questions

**Comments to the Author**

1. If the authors have adequately addressed your comments raised in a previous round of review and you feel that this manuscript is now acceptable for publication, you may indicate that here to bypass the “Comments to the Author” section, enter your conflict of interest statement in the “Confidential to Editor” section, and submit your "Accept" recommendation.

Reviewer #1: (No Response)

Reviewer #2: All comments have been addressed

2. Does this manuscript meet PLOS Global Public Health’s publication criteria? Is the manuscript technically sound, and do the data support the conclusions? The manuscript must describe methodologically and ethically rigorous research with conclusions that are appropriately drawn based on the data presented.

Reviewer #1: Yes

Reviewer #2: Yes

3. Has the statistical analysis been performed appropriately and rigorously?

Reviewer #1: No

Reviewer #2: N/A

4. Have the authors made all data underlying the findings in their manuscript fully available (please refer to the Data Availability Statement at the start of the manuscript PDF file)?

Reviewer #1: Yes

Reviewer #2: Yes

5. Is the manuscript presented in an intelligible fashion and written in standard English?

Reviewer #1: Yes

Reviewer #2: Yes

6. Review Comments to the Author

Reviewer #1: Analytical strategy:

Issues with bivariate and multivariate analysis

The decision to perform Chi square and T-test to determine the relationship between Burn out and independent variables was not properly justified at the method section and poorly presented at the result section.

Suggestion:

1. Though a score was obtained for Burn out, this was clearly a categorical outcome variable (Yes = Burn out, No = No burn out). Therefore, it was not feasible to conduct T-test for those suppose continuous variables and burn out. Best option was Chi square. In fact, the bivariate analysis could have been Chi square analysis of the outcome variables and the independent variables, including those continuous variables. Even with those continuous variables, were they parametric data? If no, then T-test cannot be the best approach.

2. After the Chi square Test, those statistically significant at this level would then be fitted for multivariate analysis.

Reviewer #2: - The comments in the introduction, results and discussion sections have been addressed.

- I am uncertain whether the last sentence in the introductory paragraph is necessary. It may be more appropriate to place it under the study design section.

- Apart from the grammatical errors identified by the other reviewer(s), the work appears to be in excellent condition for consideration at this stage.

7. PLOS authors have the option to publish the peer review history of their article (what does this mean?). If published, this will include your full peer review and any attached files.

**Do you want your identity to be public for this peer review?** For information about this choice, including consent withdrawal, please see our Privacy Policy.

Reviewer #1: No

Reviewer #2: No

---

## [Decision Letter · Decision Letter 2]

Burnout and sleep problems among nurses working in a tertiary hospital in Kathmandu, Nepal

PGPH-D-24-02324R2

Dear Mx Kanak,

We are pleased to inform you that your manuscript 'Burnout and sleep problems among nurses working in a tertiary hospital in Kathmandu, Nepal' has been provisionally accepted for publication in PLOS Global Public Health.

Best regards,

Julia Robinson

Executive Editor

Reviewer Comments (if any, and for reference):

Reviewer's Responses to Questions

**Comments to the Author**

1. If the authors have adequately addressed your comments raised in a previous round of review and you feel that this manuscript is now acceptable for publication, you may indicate that here to bypass the “Comments to the Author” section, enter your conflict of interest statement in the “Confidential to Editor” section, and submit your "Accept" recommendation.

Reviewer #1: All comments have been addressed

2. Does this manuscript meet PLOS Global Public Health’s publication criteria? Is the manuscript technically sound, and do the data support the conclusions? The manuscript must describe methodologically and ethically rigorous research with conclusions that are appropriately drawn based on the data presented.

Reviewer #1: Yes

3. Has the statistical analysis been performed appropriately and rigorously?

Reviewer #1: Yes

4. Have the authors made all data underlying the findings in their manuscript fully available (please refer to the Data Availability Statement at the start of the manuscript PDF file)?

Reviewer #1: Yes

5. Is the manuscript presented in an intelligible fashion and written in standard English?

Reviewer #1: Yes

6. Review Comments to the Author

Reviewer #1: (No Response)

7. PLOS authors have the option to publish the peer review history of their article (what does this mean?). If published, this will include your full peer review and any attached files.

**Do you want your identity to be public for this peer review?** For information about this choice, including consent withdrawal, please see our Privacy Policy.

Reviewer #1: No
